# Pedagogical strategies for teaching nursing theories in undergraduate students: A scoping review protocol

Aurélie Demagny-Warmoes[1,2], Paul Quindroit[3], Sébastien Colson[1]*

1 Faculty of Medical and Paramedical Sciences, Aix Marseille University, APHM, CEReSS, Nursing School, Marseille, France, 2 Nursing School, Valenciennes General Hospital, 59300, Valenciennes, France, 3 University Lille, CHU Lille, ULR 2694 - METRICS: Évaluation des Technologies de Santé et des Pratiques Médicales, F-59000, Lille, France

* sebastien.colson@ap-hm.fr

**Data Availability Statement:** For your information, the raw data has been deposited in the Open

## Abstract

From a pedagogical point of view, there is a great deal of creativity and practice in teaching conceptual models and nursing theories. However, there seems to be no synthesis of knowledge regarding the pedagogical methods used to teach them. The purpose of this scoping review is to map the available literature on the teaching of nursing conceptual models and theories in undergraduate nursing education. The scoping review will be developed in accordance with the JBI scoping review methodology and the PRISMA scoping review checklist. The aim of the results is to map the available literature data that can serve as inspiration and a foundation for the development of specific courses on conceptual models and theories in nursing sciences. This scoping review will be served as the basis for a study that will be described and explored the integration of nursing knowledge into bachelor of nursing curricula.

## Introduction

Since the mid-twentieth century, nursing has developed a significant body of scientific knowledge, organized and composed of conceptual models and theories developed from research [1]. These elements constitute the substance of the discipline and, ipso facto, of nursing practice. Conceptual models and theories in nursing explain the problems of individuals, promote health, recommend clinical reasoning approaches and the implementation of evidence-based interventions [2]. Although nursing knowledge is available, there are reports of minimal use of nursing theory to guide nursing practice and more frequent use of disciplinary borrowing [3]. However, nursing theories help to better understand phenomena, structure nurses' activities, and guide their practice [1]. Indeed, with the implementation of a theoretical framework, nursing interventions possess the capability to have a real impact on the quality of care, with measurable spin-offs. Conversely, a lack of theoretical support leads to less effective care interventions and a lack of argumentation regarding the rationale for these interventions and their impact on health [3].

Science Framework, accessible at the following link: https://osf.io/9naxp/

**Funding:** Hôpitaux Universitaires de Marseille (APHM) have converted the publication cost.

**Competing interests:** The authors have declared that no competing interests exist.

Therefore, in order to bring about a unique attitude and point of view, nursing interventions must be grounded in theoretical knowledge that is exclusive to their discipline [4]. From this premise, the utilization of theoretical frameworks from nursing science in teaching becomes essential to demonstrate nursing's distinctive contribution to health care. For the pioneers of disciplinary construction, subsequent generations of nurses must reap the benefits of this pedagogical approach. Thus, they designed educational programs based on a theoretical foundation that constituted a reference system for university instruction, ultimately leading to the academicization of the field [4]. The systematic introduction of care theories into nursing education curricula seems to enhance the value of teaching, promote the adoption of theory as a foundational component, and prepare students to apply it in dynamic and increasingly complex clinical environments [5]. However, the reduction or exclusion of such content within nursing curricula hinders the acquisition of the cognitive skills essential for students for identifying the theoretical and scientific elements crucial in nursing practice [5].

Incorporating the theoretical and philosophical foundations of the discipline into teaching makes the latter a powerful vector for disseminating and promoting professional values and leads to a better appropriation of nursing knowledge [6]. Therefore, an education program that integrates theory alongside critical reflection of thoses theories stand as a cornerstone for the ongoing progress of the profession and the preservation of the discipline of nursing [7].

Hence, the American Association of Colleges of Nursing (AACN), within its updated Essentials for Professional Nursing Education, acknowledges the importance of integrating theories of care into educational programs. It advocates for a theoretical approach to steer students' academic readiness [8]. This perspective aligns with Fawcett's viewpoint, outlining a blueprint for incorporating conceptual frameworks and/or nursing theories into training programs [9]. Fawcett's guideline encourages educational institutions to leverage nursing conceptual models and/or theories as foundational elements, influencing institutional project, objectives and pedagogical sequences.

There is a wealth of innovative and practice in teaching conceptual models and nursing theories in education [10]. However, the existing literature on this subject seems to be scattered. Therefore, it appears essential to map the available data in the literature that could serve as a source of inspiration and as a basis for the construction of specific courses on the fundamentals of care in initial nursing education.

## Scoping review objectives

The purpose of this scoping review is to i) identify existing pedagogical approaches used in education nursing conceptual models and/or theories in undergraduate education, ii) describe how these teachings are structured, iii) report on the outcomes of these interventions within clinical practice, iiii) identify the most commonly taught nursing conceptual models or theories.

## Review questions

The key question of the scoping review is: What pedagogical strategies are used to teach conceptual models and/or theories of nursing in undergraduate education?

The specific questions are:

Does the instruction of nursing conceptual models and/or theories occur sporadically within a singular teaching unit, or is it part of a systematically interdisciplinary, and progressive educational approach?

Does the teaching predominantly rely on a singular nursing model/theory or involve a blend of multiple nursing models/theories?

Does the teaching primarily follow a singular pedagogical practice, or does it incorporate a blend of multiple pedagogical practices?

Which nursing conceptual models or theories are most commonly taught in initial education?

What impact do these courses have on nursing practice in clinical settings and environments?

Is the program based on a nursing conceptual model/theory?

## Methods

Synthesizing scientific knowledge is one of the most important aspects of a researcher's work. There are several methods on the literature review continuum, rapid review, narrative review, integrative review, scoping review, systematic review, etc. Since each has different goals, they should not compete with each other, but rather complement each other [11]. Indeed, scoping reviews are more appropriate for broad questions and complex topics. They are an effective way to survey the literature when it is highly heterogeneous, especially in terms of methods. They help to identify and map literature data to guide practice within a specific domain. Indeed, this type of knowledge synthesis must be rigorous, transparent, and reproducible. It must reflect the values of research and researchers. The scoping review is the suitable methodology to achieve the objective of this research, which is to map existing pedagogical strategies for teaching conceptual models and/or nursing theories. The review will be conducted in accordance with Chapter 10 of the Joanna Briggs Institute (JBI) Manual for Evidence Synthesis: Scoping Reviews [12]. Results will be reported according to the preferred reporting elements for systematic reviews and the extension of meta-analyses for PRISMA ScR scoping reviews.

## Protocols and registrations

A preliminary search of PROSPERO, MEDLINE, the Cochrane Database of Systematic Reviews and the JBI Evidence Synthesis was conducted on September 1, 2023. No recent scoping or systematic reviews on this topic were found.

To ensure scientific rigor and transparency throughout the research process and to optimize the publication of the scoping review upon completion, the protocol must be recorded in an ad hoc database. The scoping review protocol was registered in advance with the Open Science Frameworks (OSF) on December 08, 2023.

## Inclusion criteria

In accordance with the recommendations of Peters et al the Population Concept Context (PCC) tool was used to identify the focus and context of the review and to design the literature search strategy [12]. The PCC tool was used to identify the focus and context of the review, and to design the literature search strategy.

### Population

The eligible population includes students enrolled in a first cycle nursing program. Many terms are used to describe initial nursing education. Therefore, all articles using terms related to nursing students and/or the first cycle level of education, such as baccalaureate, undergraduate, bachelor, student nursing, *etc*., will be included. Articles focusing solely on students at the master's (second cycle), doctoral (third cycle), or continuing education level will be excluded.

## Concept

The concept of interest is the teaching of conceptual models and theories of nursing in undergraduate education. This systematic scoping review will focus on literature that explicitly describes pedagogical strategies used to teach conceptual models and/or theories derived exclusively from nursing. All conceptual models and theories, regardless of breadth or specificity, will be included. However, conceptual frameworks from other disciplines, such as education, sociology, anthropology, *etc.* will be excluded. Additionally, literature advocating for teaching conceptual models and/or theories of care in initial training programs without explicit details on pedagogical strategies will also be excluded.

## Context

This review will examine articles that report pedagogical strategies used by educators to acculturate pre-service students with nursing conceptual models and theories. The scope encompasses education and includes pre-service nursing programs that integrate in their curriculum the teaching of nursing conceptual models and/or theories without geographic or cultural limitations. These settings can vary from the classroom to the nursing unit. Moreover, the teaching interventions may occur at any point within the first cycle curriculum. The first-year cycle nursing education includes nursing education programs internationally that are prerequisites for registration or licensure in colleges or universities.

## Information sources

The review will include qualitative, quantitative, and mixed methods study designs. Observational studies, case studies, and narrative case reports will also be included. In addition, guidelines, expert/theorist discussions, essays, letters to the editor, op-eds and articles, and gray literature that explore strategies for teaching conceptual models and/or nursing theories in pre-service education will be considered. In short, the review remains open to the inclusion of any relevant document that could enhance the dataset, without limitation on publication dates.

To minimize the risk of misinterpretation, only articles published in English and French (the native language of the principal investigator) will be included. Although excluded, all articles that do not meet the language inclusion criteria will be recorded and identified in the flowchart.

## Search strategies

The research strategy will follow the three-step process recommended by JBI [12]. The first step is to conduct a limited search in MEDLINE (PubMed) and CINAHL (EBSCO) to identify articles related to the topic. An analysis of the words contained in the titles and abstracts of these articles, as well as the index terms used, will be carried out. In the second stage, the vocabulary and syntax (keywords and index terms identified in PubMed and CINAHL) will be adapted to query other electronic databases. Finally, the reference lists of the included documents will be examined and a Web Of Science (WOS) query will be made based on the PubMed-Indexed for MEDLINE (PMID) of said documents in order to search for additional articles. This method mirrors a "snowball" search, as termed in scientific contexts.

The databases queried for this review will be: MEDLINE (PubMed); CINAHL and ERIC (via EBSCO), EMBASE, WOS. Sample search strategy for PubMed and CINAHL are outlined in Tables 1 and 2, respectively. To guarantee the most exhaustive literature search possible, sources from the reference analysis of included articles and grey literature must incorporated. Grey literature will therefore be searched using University Documentation System (SUDOC), Hyper Article online (HAL), ProQuest Dissertations and Theses Global and Google Scholar.

**Table 1. PubMed search equation strategy.**

| | Equation PubMed | |
|---|---|---|
| #1 | (nursing theory[MeSH Major Topic]) OR (models,nursing[MeSH Terms]) | 15 308 |
| #2 | ("nursing theor*"[Title/Abstract]) OR ("nursing model*"[Title/Abstract]) OR (theory-guided practice[Title/Abstract]) OR (theorist*[Title/Abstract]) OR (conceptual nursing model*[Title/Abstract]) | 5 738 |
| #3 | #1 OR #2 | 19 470 |
| #4 | ("Students, Nursing"[Mesh]) OR (((education, nursing,baccalaureate[MeSH Terms]) OR (education, nursing, diploma programs[MeSH Terms])) OR (nursing education research[MeSH Terms]) OR "Education, Nursing"[Mesh:NoExp]) | 75 571 |
| #5 | ((((baccalaureate nursing education[Title/Abstract])) OR (nursing education[Title/Abstract])) OR (nursing baccalaureate[Title/Abstract])) OR (nursing students[Title/Abstract]) | 33 619 |
| #6 | #4 OR #5 | 84 575 |
| #7 | teaching,methods[MeSH Terms] | 93 389 |
| #8 | ("teaching method*"[Title/Abstract] OR "teaching strateg*"[Title/Abstract] OR "teaching theor*"[Title/Abstract]) | 10 794 |
| #9 | #7 OR #8 | 100 075 |
| #10 | #6 OR #9 | 170 246 |
| #11 | #3 AND #10 | 3 079 |

To ensure transparency, all equation tests, keyword tests, *etc.* in all databases will be recorded and documented in an Excel spreadsheet. The detailed description of the search strategies in the selected databases will allow their replication if necessary.

## Selecting sources of evidence

Once the specific databases have been queried according to predefined protocol, all identified sources will be gathered and exported to Zotero 6.0.27/2023. Subsequently, duplicate entries will be eliminated. After weeding, the sources will be exported to the JBI system for unified information management, evaluation and review (JBI SUMARI) (JBI, Adelaide, Australia)

**Table 2. CINAHL search equation strategy.**

| | Equation CINAHL | |
|---|---|---|
| S1 | (MM "Nursing Theory+") | 7 300 |
| S2 | (MH "Education, Nursing, Theory-Based") | 260 |
| S3 | (S1 OR S2) | 7 474 |
| S4 | (TI "nursing theor*" OR AB "nursing theor*" OR TI "nursing model*" OR AB "nursing model*" OR TI "conceptual nursing model*" OR AB "conceptual nursing model*" OR TI "theory-guided practice" OR AB "theory-guided practice") | 3 647 |
| S5 | (S3 OR S4) | 9 623 |
| S6 | (MH "Students, Nursing, Baccalaureate+") | 6 649 |
| S7 | (MH "Education, Nursing, Baccalaureate+") | 12 747 |
| S8 | (MH "Education, Nursing, Research-Based") | 93 |
| S9 | (S6 OR S7 OR S8) | 16 815 |
| S10 | TI ((nurs* N2 (student* OR undergraduate OR baccalaureate OR education))) OR AB ((nurs* N2 (student* OR undergraduate OR baccalaureate OR education))) | 67 320 |
| S11 | (S9 OR S10) | 73 883 |
| S12 | (TI "teaching strateg*" OR AB "teaching strateg*" OR TI "teaching method*" OR AB "teaching method*" OR TI "teaching theor*" OR AB "teaching theor*") | 6 716 |
| S13 | (S11 OR S12) | 77 999 |
| S14 | (S5 AND S13) | 1 112 |

[13]. After conducting pilot testing, titles and abstracts will be reviewed and compared to the study eligibility criteria. Selected sources will be retrieved in full and evaluated in detail against the inclusion criteria. Reasons for excluding sources read in their entirety will be recorded and reported in the review. The selection (first sorting) and evaluation (second sorting) of sources will be carried out by two independent reviewers. Any discrepancies between the reviewers at any stage of the process will be resolved by discussion with an additional reviewer. The results of the search and study inclusion process will be fully reported in the final version of the scoping review and presented in a PRISMA flowchart.

## Data extraction

Data will be extracted by two independent reviewers using a data extraction tool developed by the research team. The development of this tool is based on the writings of Raynal & al that model the design of a pedagogical strategy [14]. Extracted data will be organized within an Excel spreadsheet and will include, firstly, article-specific characteristics: title, author, year of publication, type of publication, geographical location, etc. Secondly, a transcript of the main relevant findings that shed light on the objectives and answer the review questions will be compiled: context, teaching techniques, didactic material, group size, performance level, conceptual models and/or nursing theories taught, their combination, institutional support, impact of this teaching on practice, etc.

This standardized data extraction model is tested by reviewers at the beginning of the article analysis process. Given the iterative nature of the data extraction process, the model will be modified, revised, and refined as needed [12]. Changes will be comprehensively detailed within the scoping review. Discrepancies between reviewers will be addressed through discussion, aiming for consensus. In cases where consensus cannot be achieved, resolution will involve the intervention of a third party. Authors of articles will be contacted, as appropriate, to solicit missing. The process of resolving disagreements and the evidence selection procedure will be described in narrative and/or flowchart form [15].

## Data analysis

The aim of this review is to provide an overview of the state of the art of existing pedagogical strategies for teaching conceptual models and/or theories with undergraduate nursing education. Unlike systematic literature reviews, this review does not intend to develop guidelines. Therefore, the analysis of the data does not aim to assess the quality of the evidence, the completeness of the source, or potential biases, but rather to explore how, by whom, and for what purpose the teaching of conceptual models and theories of care is implemented [12]. To meet to the standards of the scoping review, the analysis of sources will be qualitative and descriptive in nature, providing a logical summary that is consistent with the objectives of the review [12]. The extraction tool will allow, among other things, to code and classify the different types of pedagogical interventions, the different theorists addressed and their frequency.

## Results presentation

Data will be mapped and presented in schematic or tabular formats to answer research questions [12]. Alongside these visual representations, a narrative summary will complement the results, describing how the results relate to the review objective and questions. Subsequently, the results will undergo discussion, highlighting any voids or constraints evident within the existing literature.

## Limits

A common limitation of this type of review is the potential omission of relevant studies [16]. To limit this selection bias, the systematic approach of scoping reviews stands as a valuable alternative [17]. Consequently, the systematization of this review will be based on the PRISMA ScR and JBI SUMARI guidelines. Moreover, the protocol has been deposited in the Open Science Framework, facilitating the identification and reporting of any discrepancies between the prescribed protocol and the actual research process [12].

Again, to limit selection bias and to avoid missing important data, a wide range of databases aligned with the research objectives, as well as gray literature, will be consulted. Although the number of databases consulted is relatively correct, other databases may also be relevant. As for the gray literature, most of it will be dissertations. Thus, the choice of databases and gray literature may be a limitation of the review. However, the systematic approach and access to the CINAHL and PubMed database search strategy can mitigate concerns regarding of the non-exhaustiveness of the literature search. While the importance of consulting multiple databases is recognized, the question of how many sources should be reviewed to ensure a quality review is currently unresolved [18]. It is therefore difficult to know whether the number of inclusions will be sufficient to meet the research objectives.

Also, only sources published in English and French are included. For some authors, this selection criterion is seen as a limitation [19].

Finally, while emphasizing the significance of currency in scholarly articles, Pautasso prompts us to acknowledge the value of older publications, whose contributions to the field remain indispensable [20]. This perspective holds particular relevance within our journal's context, where a significant focus on teaching the discipline's fundamentals emerged around the 1970s. Thus, in an effort to mitigate selection bias, the decision was made not to impose a specific time restriction on the inclusion of sources.

## Conclusion

It's not always easy to teach conceptual models and theories of nursing care, especially in countries where the discipline of nursing is only just beginning to emerge. With a view to sharing, this scoping review will provide an inventory of the vast majority of teaching strategies used to teach conceptual models and nursing theories. As such, it will provide support for teachers wishing to introduce, innovate or perfect courses on nursing theories. In addition, it will be interesting to identify teaching strategies that have been rigorously tested and evaluated in the light of research methods. The scoping review is part of a dissertation, one of the main aims of which is to find ways to support the teaching of disciplinary fundamentals.

## Supporting information

**S1 Checklist. PRISMA-P (Preferred Reporting Items for Systematic review and Meta-Analysis Protocols) 2015 checklist: Recommended items to address in a systematic review protocol\*.** (DOCX)

## Author Contributions

**Conceptualization:** Aurélie Demagny-Warmoes, Paul Quindroit, Sébastien Colson.

**Investigation:** Aurélie Demagny-Warmoes.

**Methodology:** Aurélie Demagny-Warmoes, Paul Quindroit, Sébastien Colson.

**Project administration:** Aurélie Demagny-Warmoes.

**Resources:** Aurélie Demagny-Warmoes, Sébastien Colson.

**Supervision:** Aurélie Demagny-Warmoes, Paul Quindroit, Sébastien Colson.

**Validation:** Paul Quindroit, Sébastien Colson.

**Writing – original draft:** Aurélie Demagny-Warmoes.

**Writing – review & editing:** Aurélie Demagny-Warmoes, Paul Quindroit, Sébastien Colson.

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
