## [Decision Letter · Decision Letter 0]

1 Jul 2024

PONE-D-24-16045Pedagogical strategies for teaching nursing theories in undergraduate students: a scoping review protocolPLOS ONE

Dear Dr. DEMAGNY - WARMOES

Thank you for submitting your manuscript to PLOS ONE. After careful consideration, we feel that it has merit but does not fully meet PLOS ONE’s publication criteria as it currently stands. Therefore, we invite you to submit a revised version of the manuscript that addresses the points raised during the review process.

We look forward to receiving your revised manuscript.

Kind regards,

Maria José Nogueira, Ph.D.

Academic Editor

PLOS ONE

Journal Requirements:

2. We note that your Data Availability Statement is currently as follows: "All relevant data are within the manuscript and its Supporting Information files."

Additional Editor Comments:

Please consider:

to revise the use of Keywords vs Descriptors

Review the tables - include the Title in each one, according to Plos One standards.

Revise JBI referencies for more recent (2024)

Reviewers' comments:

Reviewer's Responses to Questions

**Comments to the Author**

1. Is the manuscript technically sound, and do the data support the conclusions?

Reviewer #1: Yes

2. Has the statistical analysis been performed appropriately and rigorously? 

Reviewer #1: N/A

3. Have the authors made all data underlying the findings in their manuscript fully available?

Reviewer #1: Yes

4. Is the manuscript presented in an intelligible fashion and written in standard English?

Reviewer #1: Yes

5. Review Comments to the Author

Reviewer #1: Dear authors and editors,

Congratulations on your manuscript, it contains new ideas and findings and makes new contributions to the literature, presenting an original and relevant topic.

Please consider to review the keyword: "scoping review protocol", because it is not MESH term.

Please consider to check:

- Aromataris E, Lockwood C, Porritt K, Pilla B, Jordan Z, editors. JBI Manual for Evidence Synthesis. JBI; 2024. Available from: https://synthesismanual.jbi.global. https://doi.org/10.46658/JBIMES-24-01

- Peters MDJ, Godfrey C, McInerney P, Munn Z, Tricco AC, Khalil, H. Scoping Reviews (2020). Aromataris E, Lockwood C, Porritt K, Pilla B, Jordan Z, editors. JBI Manual for Evidence Synthesis. JBI; 2024. Available from: https://synthesismanual.jbi.global. https://doi.org/10.46658/JBIMES-24-09.

Many congratulations on your manuscript and huge success for publication and future studies.

Sincerely

6. PLOS authors have the option to publish the peer review history of their article (what does this mean?). If published, this will include your full peer review and any attached files.

Reviewer #1: No

---

## [Author Response · Author response to Decision Letter 0]

28 Jul 2024

-Point-by-point response letter-

Response to Editor 

We thank Editor for comments and constructive suggestions. We have carefully revised the manuscript according to each comment and suggestion. 

Point 1: PLOS ONE's style requirements 

Line: 1,2,3,16,57,63,78,91,98,124,134,165,176,191,201,206,227,239.

All titles have been formatted in Times New Roman size 18, and the subtitles in Times New Roman size 16. Line: 102,108,116.

Line: 58,64,79,92,99,103,109,117,125,135,166,192,202,207,228. An indent has been added at the beginning of each paragraph.

Line: 153, 162. The titles of tables 1 and 2 now comply with the editorial guidelines.

Line: 4 to 13. The abstract has been written in accordance with the editor's recommendations.

Point 2: Data Availability Statement

This protocol has been registered on the Open Science Framework website. It is currently under restricted access until the article is published. Here is the registration link: 

http://osf.io/gj35n

Point 3: Billing option

The manuscript has been revised to change the corresponding author. The new corresponding author is Professor Sébastien Colson at the following address: sebastien.colson@ap-hm.fr. Professor Colson, the last author of this article, is affiliated with APHM, which subsidizes the publication fees. Affiliations 2 and 3 have been modified as well. 

Point 4: Information files 

No supplementary information file

Point 5: Reference list

As requested by the reviewer, references 12 and 15 have been modified.

Reference 12 : Peters M, Gogfrey C, Mclnerney P, Munn Z, Tricco A, Khalil H. Chapter 11: Scoping reviews. In: Aromataris E, Munn Z (Editors) JBI Manuel for Evidence Synthesis, JBI [Internet]. 2020. Available from: https://synthesismanual.jbi.global. https://doi.org/10.46658/JBIMES-20-12

Line: 262 to 265. Replaced by:Peters MDJ, Gogfrey C, Mclnerney P, Munn Z, Tricco AC, Khalil H. Scoping Reviews. In: Aromataris E, Lockwood C, Porritt K, Pilla B, Jordan Z (Editors) JBI Manuel for Evidence Synthesis, JBI; 2024. Available from: https://synthesismanual.jbi.global. https://doi.org/10.46658/JBIMES-24-09

Reference 15: Peters MDJ, Marnie C, Tricco AC, Pollock D, Munn Z, Alexander L, et al. Updated methodological guidance for the conduct of scoping reviews. JBI Evid Implement. 2021 Mar;19(1):3.

Line: 272 to 275. Replaced by Aromataris E, Lockwood C, Porritt K, Pilla B, Jordan Z.JBI Manual for Evidence Synthesis.JBI.2024. https://doi.org/10.46658/JBIMES-24-01

Line: 88. Since we have included more recent references, we had to change chapter 11 to chapter 10.

Response to Reviewer 

We thank Reviewer for comments and constructive suggestions. We have carefully revised the manuscript according to each comment and suggestion.

Colour code: 

Point 1: 

Please consider to review the keyword: "scoping review protocol", because it is not MESH term.

Line: 14. Indeed, "scoping review protocol" does not exist in MeSH. We propose to remove it. Since the MeSH term "protocol" primarily represents clinical research protocols, we prefer to remove this term and replace it with the MeSH term "models, nursing" to meet the recommended number of keywords.

Point 2: 

Please consider to check:

- Aromataris E, Lockwood C, Porritt K, Pilla B, Jordan Z, editors. JBI Manual for Evidence Synthesis. JBI; 2024. Available from: https://synthesismanual.jbi.global. https://doi.org/10.46658/JBIMES-24-01

- Peters MDJ, Godfrey C, McInerney P, Munn Z, Tricco AC, Khalil, H. Scoping Reviews (2020). Aromataris E, Lockwood C, Porritt K, Pilla B, Jordan Z, editors. JBI Manual for Evidence Synthesis. JBI; 2024. Available from: https://synthesismanual.jbi.global. https://doi.org/10.46658/JBIMES-24-09

As requested by the reviewer, references 12 and 15 have been modified.

Line: 262 to 265. Replaced reference 12 by: Peters MDJ, Gogfrey C, Mclnerney P, Munn Z, Tricco AC, Khalil H. Scoping Reviews. In: Aromataris E, Lockwood C, Porritt K, Pilla B, Jordan Z (Editors) JBI Manuel for Evidence Synthesis, JBI; 2024. Available from: https://synthesismanual.jbi.global. https://doi.org/10.46658/JBIMES-24-09

Line: 272 to 275. Replaced reference 15 by: Aromataris E, Lockwood C, Porritt K, Pilla B, Jordan Z.JBI Manual for Evidence Synthesis.JBI.2024. https://doi.org/10.46658/JBIMES-24-01

---

## [Editor Report · Decision Letter 1]

31 Jul 2024

Pedagogical strategies for teaching nursing theories in undergraduate students: a scoping review protocol

PONE-D-24-16045R1

Dear Dr. Sébastien Colson,

We’re pleased to inform you that your manuscript has been judged scientifically suitable for publication and will be formally accepted for publication once it meets all outstanding technical requirements.

Kind regards,

Maria José Nogueira, Ph.D.

Academic Editor

PLOS ONE

Additional Editor Comments (optional):

The authors have made the changes recommended by the reviewers. The manuscript is now ready to be accepted for publication.

Congratulations
---

## [Editor Report · Acceptance letter]

7 Oct 2024

PONE-D-24-16045R1 

PLOS ONE

Dear Dr. Colson, 

I'm pleased to inform you that your manuscript has been deemed suitable for publication in PLOS ONE. Congratulations! Your manuscript is now being handed over to our production team.

Kind regards, 

on behalf of

Professor Maria José Nogueira 

Academic Editor

PLOS ONE